# Non-Ruptured Temporal Lobe Dermoid Cyst Concomitant with Focal Cortical Dysplasia Causing Temporal Lobe Epilepsy—A Case Report and Literature Review

**DOI:** 10.3390/brainsci11091136

**Published:** 2021-08-27

**Authors:** Keisuke Hatano, Ayataka Fujimoto, Chikanori Inenaga, Yoshiro Otsuki, Hideo Enoki, Tohru Okanishi

**Affiliations:** 1Comprehensive Epilepsy Center, Seirei Hamamatsu General Hospital, 2-12-12 Sumiyoshi, Nakaku, Hamamatsu, Shizuoka 430-8558, Japan; hatakenosuke@gmail.com (K.H.); enokih.neuropediatr@gmail.com (H.E.); t.okanishi@tottori-u.ac.jp (T.O.); 2Department of Neurosurgery, Seirei Hamamatsu General Hospital, 2-12-12 Sumiyoshi, Nakaku, Hamamatsu, Shizuoka 430-8558, Japan; inenaga@sis.seirei.or.jp; 3Department of Pathology, Seirei Hamamatsu General Hospital, 2-12-12 Sumiyoshi, Nakaku, Hamamatsu, Shizuoka 430-8558, Japan; otsuki@sis.seirei.or.jp

**Keywords:** dermoid cyst, intramedullary, epileptic seizure, non-ruptured, focal cortical dysplasia

## Abstract

Background: Intracranial dermoid cyst is a rare, benign, nonneoplastic tumor-like lesion that could cause seizures, headache, and hydrocephalus. We hypothesized that the temporal lobe dermoid cyst in combination with other factors were causing the epileptic seizure. Methods: We encountered a 17-year-old girl with anti-seizure medication-resistant epilepsy secondary to dermoid cyst located in the temporal region depicted on magnetic resonance imaging (MRI). She showed neither symptoms of meningitis nor rupture of the cyst according to serial MRI. We hypothesized that temporal lobe dermoid cyst in combination with other factors, such as focal cortical dysplasia (FCD), etc., was causing epileptic seizures in this case. She underwent dermoid cyst removal surgery with resection of the tip of the antero-inferior temporal lobe. Results: Histopathological study showed multiple small intramedullary dermoid cysts in the left antero-inferior temporal lobe in addition to MRI lesions and FCD. Conclusion: A patient with medically intractable epilepsy secondary to left temporal lobe dermoid cyst showed multiple intramedullary dermoid cysts and focal cortical dysplasia that might have interacted to create epileptogenicity. To our knowledge, this is the first case report of dermoid cyst concomitant with FCD.

## 1. Introduction

Intracranial dermoid cyst is a rare, benign, nonneoplastic tumor-like lesion representing 0.04–0.6% of primary brain tumors [1]. Even though both dermoid cyst and teratoma can contain hair or bone components, dermoid cyst is considered to arise from ectodermal cells, whereas teratoma is considered to arise from misplaced multipotent germ cells [2,3,4,5]. Dermoid cysts are different from teratoma, which is classified as a neoplastic tumor [6] containing all three embryonic germ cell layers [3]. Dermoid cysts are also distinguishable from epidermoid cysts, which contain keratinaceous debris, solid crystalline cholesterol, but not dermal appendages, even though both are congenital, ectodermal inclusion cysts [7].

Clinically, seizure (44%) and headache (67%) are common [5,8]. Two types of seizures have been described in previous reports. One type involves generalized seizures that are considered to result from exposure to inflammatory contents following rupture of the dermoid cyst [7,8,9,10,11,12]. Generalized seizures concomitant with symptoms of aseptic meningitis are explained by this theory and are thus considered an acute symptomatic seizure. The other type of seizure involves impaired awareness. Many reports have described patients showing not only the type of generalized seizure with symptoms of aseptic meningitis, but also chronic impaired awareness seizures without concomitant aseptic meningitis, which could be considered as epileptic seizures [7,8,9,10,11].

We encountered a patient with anti-seizure medication (ASM)-resistant epilepsy with a dermoid cyst located in the temporal region, who never showed symptoms of aseptic meningitis. We hypothesized that the temporal lobe dermoid cyst in combination with other factors such as focal cortical dysplasia (FCD), etc., were causing the epileptic seizure. 

## 2. Case Presentation

### 2.1. Clinical Course

A 17-year-old left-handed girl was referred from a local physician for control of medically intractable epileptic seizures. She had experienced epileptic seizures since the age of 13 years. At that time, she had described a weekly olfactory aura as an indescribable but uncomfortable smell, followed by epigastric sensation. Those seizures often (monthly) led to loss of awareness with mouthing, drooling, and dystonic posturing of the right hand. This impaired awareness seizure generally lasted 1–2 minutes. This impaired awareness seizure rarely proceeded to bilateral tonic-clonic seizures, only once in several years. Electroencephalograms (EEG) showed left temporal focal slowing without epileptiform discharges (Figure 1). When she underwent long term video EEG for a week, she only reported the olfactory aura without impaired awareness seizure, and showed no EEG changes. Magnetic resonance imaging (MRI) of the brain showed a left temporal lesion. Left temporal lobe epilepsy secondary to the left temporal lesion was diagnosed and levetiracetam was started. The patient remained free from focal-onset impaired awareness seizures and focal-to-bilateral tonic-clonic seizures for three years, but still experienced olfactory auras during this period.

Olfactory auras gradually increased in frequency to daily and also increased in intensity. Focal-onset impaired awareness seizures and focal-to-bilateral tonic-clonic seizures also recurred from the age of 17 years. Lacosamide was added to levetiracetam, but seizures were not controlled. She was therefore referred to our hospital for consideration of the possibility of epilepsy surgery.

No neurological abnormalities were identified other than the epileptic seizures. Laboratory data were within normal ranges, including alpha fetoprotein and human chorionic gonadotropin.

### 2.2. Neuropsychological Examinations

The Wechsler adult intelligence scale-4th edition showed: total intelligence quotient (IQ) index, 114; verbal comprehension index, 110; perceptual reasoning index, 114; working memory index, 109; and processing speed index 118. The revised Wechsler memory scale showed: auditory memory index, 129; visual memory index, 111; general memory index, 129; attention index, 121; and delayed memory index, 124.

To perform Wada test, we placed the 4-French catheter in the internal carotid artery through a guiding catheter in the right femoral artery. We manually injected diluted propofol (10 mg per 10 mL solution of saline) through the catheter. Scalp EEG was performed simultaneously to observe the diffuse slowing produced by the propofol in the injected hemisphere [13]. The Wada test showed left-hemisphere language dominance.

### 2.3. Neuro-Imaging

MRI (Figure 2) showed a well-delineated lesion in the left temporal lobe. The lesion had two major components. One component was mainly located in the medial temporal region and showed signal heterogeneity on T1-weighted imaging (WI) and hyperintensity on T2-WI. This massive component was displacing the left temporal lobe upwards and the hippocampus backward. This massive component was an extra-axial lesion, because T1-WI and fluid-attenuated inversion recovery (FLAIR) imaging showed a thin, hypointense space between the lesion and temporal lobe structures. The other component was a cystic part at the bottom of the temporal tip. This cystic part existed inside the temporal lobe. However, computed tomography showed a high-density lesion in the temporal tip, continuous with the massive medial temporal lesion. Diffusion-weighted imaging showed a low-intensity lesion with a slight hyperintense area inside the lesion.

### 2.4. Surgery

Since seizures had remained uncontrolled, she exhibited both right hand dystonic posturing during impaired awareness seizures [14], and focal slowing in the left temporal region on EEG, we regarded those seizures as associated with the left temporal lesion. We therefore performed left temporal lesionectomy and resection of the tip of the antero-inferior temporal lobe in an awake state. We limited the resection area to the antero-inferior temporal lobe with the lesion because the patient showed high IQ and memory scores. We therefore designed the resection area from the beginning without performing a subdural electrode study in an attempt to preserve these higher brain and memory functions.

The inferior temporal gyrus was slightly firm. We first approached the lesion from the superior side of the firm inferior temporal gyrus. We then approached the massive cystic part, which was encapsulated by a thin, transparent, and fibrous membrane. Inside the capsule, yellowish, scrambled-egg-like creamy parts were found, containing many white, nose-hair-like structures. We completely removed this creamy lesion with its fibrous capsule, which was not adherent to surrounding structures and was easily removable. The cystic part in the anterior temporal tip was also removed with the antero-inferior tip of the temporal lobe. Neither dermal sinus nor bone deficits were seen.

### 2.5. Histopathology

The massive cystic lesion depicted in the MRI contained keratinaceous debris. Other small cystic structures were identified in the brain parenchyma. These small cystic structures had an epithelial lining of keratinizing stratified squamous epithelium and were seen in the cortex. Some parts of the epithelium of the cysts contained born components. Hairs were seen adjacent to the bone components and also in the brain parenchyma. The keratinized substance was present in a form that was involved in the brain parenchyma, surrounded by glial cell proliferation. Near the keratinized substance, dysmorphic neurons situated in the white matter were indicative of focal cortical dysplasia (FCD) (Figure 3).

### 2.6. Postoperative Course

Following excision of the tip of the left temporal lobe, which included the cystic lesion, the patient remained free from daily olfactory auras as well as focal-onset impaired awareness seizures and focal-to-bilateral tonic-clonic seizures for more than 50 days. Post-operative MRI depicted total removal of the lesion with the tip of the left temporal lobe (Figure 4). Post-operative scalp EEG showed no focal slowing and epileptiform discharges observed, even though a continuous breach rhythm was seen in the left fronto-temporal area.

The patient remained free from daily olfactory auras right after the surgery, but as the follow-up period was short, we considered that freedom from seizures might not have been completely secured for the patient, and long-term follow-up remains important.

## 3. Discussion

We have reported a case of temporal lobe dermoid cyst concomitant with FCD that caused temporal lobe epilepsy. The dermoid cyst existed in the left temporal fossa, but not in midline structures, and the patient had never experienced any episodes of meningitis or shown any findings of rupture on serial MRI. We therefore considered that the epileptic seizures had not involved rupture of the dermoid cyst. Epileptic seizures were naturally considered to potentially involve the interplay of the FCD and dermoid cysts in the brain parenchyma [15,16,17,18].

Histopathological examination showed multiple small dermoid cysts in addition to the major dermoid component seen on MRI. FCDs and bone components and hair were found to be involved in the brain parenchyma. In the process of neural maturation, as the timing of neural tube formation is related to formation of both dermoid cysts and FCDs, this complex of abnormal structures in the left temporal lobe might arise at this time [8,19,20].

Other factors that might induce epileptogenicity could include possible foreign body reactions to keratinization of the dermoid cysts, metabolic interactions around the lesion, possible substantial secretions from the dermoid cyst, or micro-rupture that could not be detected on neuroimaging or from clinical symptoms. However, this was only a case study, and more cases need to be evaluated to clarify these factors.

However, the most important point that this case made us carefully consider was that risk factors were not the single cause of epilepsy, since the temporal lobe had both a dermoid cyst and FCD.

Incomplete resection of the dermoid cyst might cause recurrence, and gross total resection is thus expected [21,22], but since multiple dermoid cysts were present in the brain parenchyma in addition to the MRI visible lesion in this case, consideration should be given to the risk of recurrence even if gross total removal is performed, because residual dermoid cysts might be left in the brain parenchyma.

In terms of seizure, caution is also warranted not only about acute symptomatic seizures caused by ruptured dermoid cyst, but also about epileptic seizures associated with cysts located in non-midline, supratentorial regions.

For the surgical management of both dermoid cyst and ASM-resistant epilepsy, since invasive monitoring risks rupture of the dermoid cyst, detecting epileptogenicity in the brain to determine the optimal area for resection and simultaneously preserving higher brain function by brain mapping is a tall order. As this was the first case report of dermoid cyst concomitant with FCD, little is known about the exact epileptogenicity. We hope this case summaries provide valuable data. Collecting more data such as intraoperative EEG [11] in a global, multicenter study is thus warranted in future cases.

## 4. Conclusions

A patient with medically intractable epilepsy secondary to left temporal lobe dermoid cyst showed multiple intramedullary cysts and focal cortical dysplasia that might have interacted to create epileptogenicity. To our knowledge, this is the first case report of dermoid cyst concomitant with FCD.

## Figures and Tables

**Figure 1 brainsci-11-01136-f001:**
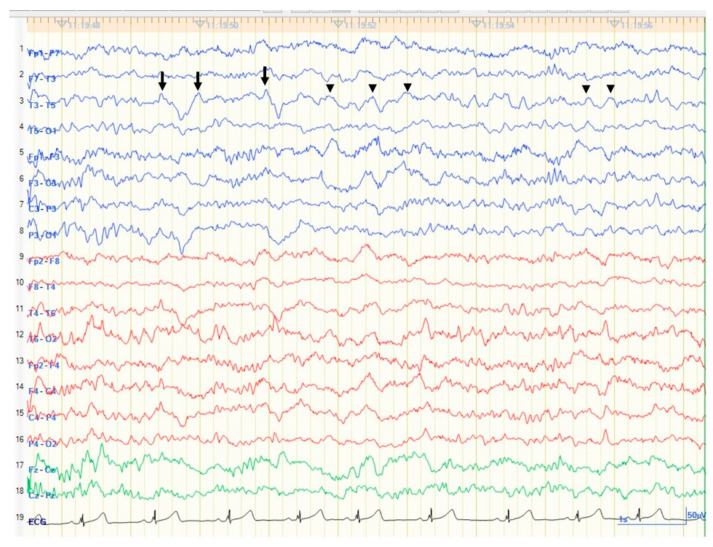
Scalp electroencephalogram of anterior–posterior bipolar montage using 10–20 International System shows medium to high amplitude sharply contoured focal slowing over the left mid-posterior temporal region (arrows). Monomorphic, medium amplitude 3–5 Hz delta and theta activities were seen in the same region (arrowheads).

**Figure 2 brainsci-11-01136-f002:**
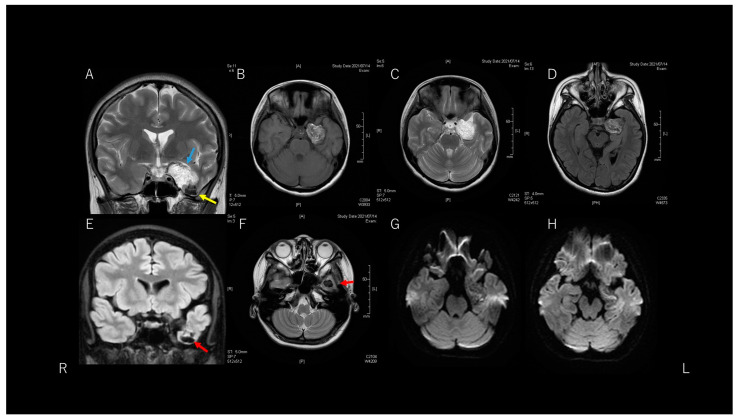
A lesion is evident in the left temporal fossa. This lesion consists of two components, indicated by sky blue and yellow arrows (**A**). Axial T1-weighted imaging (WI) and T2-WI (**B**,**C**) show heterogeneous and hyperintense parts, respectively. Since a thin space exists between the lesion and temporal lobe structures, this component is considered an extra-axial lesion. This component is displacing the temporal lobe and hippocampus (**D**). The other component is located within the temporal lobe, appearing as a low-intensity cystic lesion (**E**,**F**: red arrows). Diffusion WI (**G**,**H**) shows heterogeneous, slightly hyperintense findings.

**Figure 3 brainsci-11-01136-f003:**
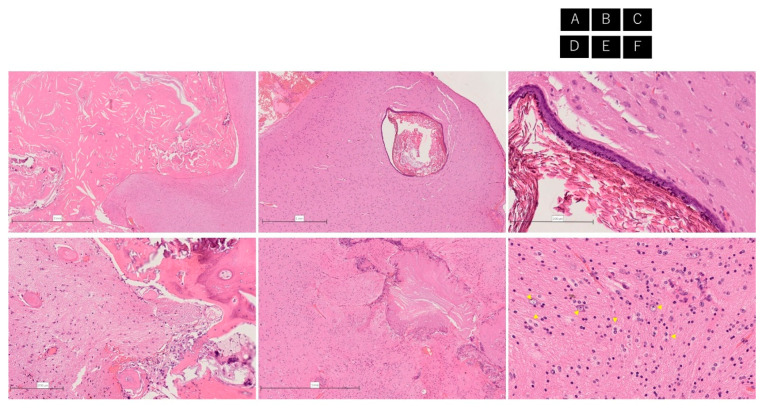
The massive cystic lesion depicted on MRI contains keratinaceous debris (**A**). In the brain parenchyma, other small cystic structures are apparent (**B**). These small cystic structures have an epithelial lining of keratinized, stratified squamous epithelium and are seen in the cortex (**C**). Some parts of the cyst epithelium contain bone components (**D**). Hairs are also seen adjacent to the bone components and in the brain parenchyma (**D**). The keratinized substance is present in a form that is involved in brain parenchyma, and surrounding glial cell proliferation is observed (**E**). Near the keratinized substance, many dysmorphic neurons (representative neurons; yellow arrowhead) are situated in the white matter. The neurons range from normal-sized to enlarged and show cytologic atypia such as abnormalities of nuclear shape, size, or nucleolar morphology, indicative of focal cortical dysplasia (**F**).

**Figure 4 brainsci-11-01136-f004:**
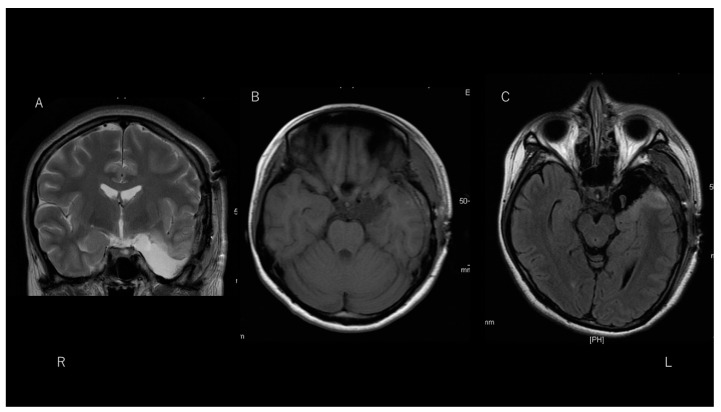
T2-weighted imaging (**A**) and T1-weighted imaging (**B**) reveal total removal of the lesion in the left temporal region. Fluid-attenuated inversion recovery imaging (**C**) of a hippocampus slice shows removal of the anterior temporal lobe tip, preserving the hippocampus.

## Data Availability

The data are not publicly available due to patients’ privacy.

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
