# Peer review of "Non-Ruptured Temporal Lobe Dermoid Cyst Concomitant with Focal Cortical Dysplasia Causing Temporal Lobe Epilepsy—A Case Report and Literature Review"

_brainsci, 2021, doi:10.3390/brainsci11091136_

Round 1

Reviewer 1 Report

The manuscript “Non-ruptured temporal lobe dermoid cyst concomitant with focal cortical dysplasia causing temporal lobe epilepsy. A case report and literature review.”, by Hatano et al, describes an interesting case of temporal lobe epilepsy. The review is well done and informative. However, there are some points that authors must address:

1.- Figure 1. Panels G and H are too small and difficult to see. Please, increase the size.

2.- Maybe to show the EEG traces should be interesting.

3.- Clarify the procedure for Wada test (propofol, amobarbital, etomidate?, EEG monitoring?).

4.- Figure 2. Please, indicate by arrow the dysmorphic neurons and show a detail.

5.- The argument for a surgical decision as a brain surgery (“she desired to be seizure free”, line 96) do not seems to be clinically founded. This reason is obvious for every patient, but a decision like this must be better justified. It is absolutely known that even in cases with macroscopic lesions, seizures can be originated in other lobes. The authors must show that seizures originated from left temporal lobe by means of EEG telemetry.

6.- It is very important to indicate the follow-up period after surgery. Besides, post-op MRi/CT and EEG must be shown.

Author Response

We greatly appreciate the thorough review and kind advice, which have helped us improve the quality of our manuscript. The manuscript has been revised to address the issues raised by the reviewers. The changes made and our point-by-point responses to the comments from the reviewer are summarized below. We have also tracked the changes in red in the revised manuscript.

1.- Figure 1. Panels G and H are too small and difficult to see. Please, increase the size.

Response:

We are sorry for the small figures. We have enlarged the figures accordingly.

2.- Maybe to show the EEG traces should be interesting.

Response:

Thank you very much for the point. We have added the EEG findings as Figure 1.

3.- Clarify the procedure for Wada test (propofol, amobarbital, etomidate?, EEG monitoring?).

Response:

We have added more detailed explanation about Wada test.

4.- Figure 2. Please, indicate by arrow the dysmorphic neurons and show a detail.

Response:

We have added some arrows for the representative neurons and added further explanation for the figure.

5.- The argument for a surgical decision as a brain surgery (“she desired to be seizure free”, line 96) do not seems to be clinically founded. This reason is obvious for every patient, but a decision like this must be better justified. It is absolutely known that even in cases with macroscopic lesions, seizures can be originated in other lobes. The authors must show that seizures originated from left temporal lobe by means of EEG telemetry.

Response:

Thank you very much for suggesting this point. We have not noticed the important point until your suggestion. Since the suggested sentences (line 96) was unscientific, we have removed the part.

We apologize for not describing the detailed seizure semiology. As we made the decision to perform the surgery based on her seizure symptoms and EEG findings, we should have added more details about her impaired awareness seizure.  

Because she underwent a long-term video EEG, we have added the explanation even though she only exhibited the olfactory auras without any EEG changes.

6.- It is very important to indicate the follow-up period after surgery. Besides, post-op MRi/CT and EEG must be shown.

Response:

We have added the follow-up period, which was 50 days. Since it might not be correct to say that the patient was free of seizures, we would like to change the phrase a little conservatively. We are sorry for not showing the post-op MRI. We have added the MRI as Figure 4. In terms of the post-op EEG, as the number of figures is limited, we described the EEG findings in the result section.

Reviewer 2 Report

  1. Overall, the manuscript is well written and the finding is very interesting not only to physicians but also to neuroscientists and biologists investigating several aspects of epilepsy. I especially appreciate the clarity with which the authors present the figures. They are very clear and well presented.
  2. The authors mention several times including in the abstract and introduction that the temporal lobe dermoid cyst in combination with other factors causes epileptic seizures. What are some examples of these factors? I think elaborating on what these ‘factors’ are, especially early on in the paper, would help set the stage for the rest of the paper.
  3. I find the second paragraph of the introduction to be rather confusing. The authors focus on generalized seizures, but also mention ‘impaired awareness seizures.’ What are the connections between these two ‘types’ in the context of this paper?
  4. Out of curiosity, have you encountered a similar case to the patient reported in the manuscript? I’m wondering how common such dysfunction exists in the general patient population.
  5. There are a few vague/unclear sentences and typos which undermine the explanation content of the manuscript. A careful reading/editing throughout would be helpful.

Author Response

We greatly appreciate the thorough review and kind advice, which have helped us improve the quality of our manuscript. The manuscript has been revised to address the issues raised by the reviewers. The changes made and our point-by-point responses to the comments from the reviewer are summarized below. We have also tracked the changes in red in the revised manuscript.

  1. The authors mention several times including in the abstract and introduction that the temporal lobe dermoid cyst in combination with other factors causes epileptic seizures. What are some examples of these factors? I think elaborating on what these ‘factors’ are, especially early on in the paper, would help set the stage for the rest of the paper.

Response:

This is an important point. Thank you very much for giving us an opportunity to mention about other factors. However, since we are unable to reply to this question from a case report, we have added some comments in the discussion section. Thank you very much again for raising this question.

  1. I find the second paragraph of the introduction to be rather confusing. The authors focus on generalized seizures, but also mention ‘impaired awareness seizures.’ What are the connections between these two ‘types’ in the context of this paper?

Response:

We are sorry for the confusion. Since we thought that it was important to include description of both types of seizures, we have explained more about generalized seizure and the impaired awareness seizure.

  1. Out of curiosity, have you encountered a similar case to the patient reported in the manuscript? I’m wondering how common such dysfunction exists in the general patient population.

Response:

This is also a very important question. The occurrence rate, as we stated in the introduction part is extremely rare.

In terms of epilepsy, only one paper (Neurol Med Chir (Tokyo) 2006, 46, 206-209, doi:10.2176/nmc.46.206) stated the relationship between the dermoid cyst and epilepsy.

However, even this paper concluded that the epileptogenicity was from the rupture of the cyst. Therefore, we hope readers of our paper will consider other mechanisms of seizures.

  1. There are a few vague/unclear sentences and typos which undermine the explanation content of the manuscript. A careful reading/editing throughout would be helpful.

Response:

We are sorry for the points.

We asked a professional English-speaking editor to check the manuscript. Since we asked the editor to make the English more natural this time, we hope this revision will satisfy reviewer 2.

Round 2

Reviewer 1 Report

The new version has been significantly improved. However, and although it is discussed at the manuscript, I have some concerns about the short follow-up period.

Author Response

Response to comments from reviewers_R2

Reviewer #1

The new version has been significantly improved. However, and although it is discussed at the manuscript, I have some concerns about the short follow-up period.

We greatly appreciate the second thorough review and helpful advice. The manuscript has been revised even though we could only follow-up a short term of the issue raised by the reviewer1.

As the most important point that we could carefully consider from this case report was that the epileptogenicity was not the single cause of the dermoid cyst but also the FCD. Therefore, we have rephrased the point accordingly. We have also removed our hypothesis from the conclusion because the most important point of this case report was that the temporal lobe had both dermoid cyst and FCD.

Once again, thank you very much for giving us valuable comments.
